# HDAC6 Negatively Regulates miR-155-5p Expression to Elicit Proliferation by Targeting RHEB in Microvascular Endothelial Cells under Mechanical Unloading

**DOI:** 10.3390/ijms221910527

**Published:** 2021-09-29

**Authors:** Liqun Xu, Lijun Zhang, Xiaoyan Zhang, Gaozhi Li, Yixuan Wang, Jingjing Dong, Honghui Wang, Zebing Hu, Xinsheng Cao, Shu Zhang, Fei Shi

**Affiliations:** 1The Key Laboratory of Aerospace Medicine, Ministry of Education, Air Force Medical University, Xi’an 710032, China; aliqunxu@fmmu.edu.cn (L.X.); 17791522721@163.com (L.Z.); zxy419452@163.com (X.Z.); ligaozhi1022@163.com (G.L.); wangyx1992@126.com (Y.W.); teamedicine@163.com (J.D.); wholefull@sohu.com (H.W.); zebinghu@fmmu.edu.cn (Z.H.); caoxinsh@fmmu.edu.cn (X.C.); 2State Key Laboratory of Space Medicine Fundamentals and Application, China Astronaut Research and Training Center, Beijing 100094, China

**Keywords:** mechanical unloading, HDAC6, miR-155-5p, RHEB, proliferation

## Abstract

Mechanical unloading contributes to significant cardiovascular deconditioning. Endothelial dysfunction in the sites of microcirculation may be one of the causes of the cardiovascular degeneration induced by unloading, but the detailed mechanism is still unclear. Here, we first demonstrated that mechanical unloading inhibited brain microvascular endothelial cell proliferation and downregulated histone deacetylase 6 (HDAC6) expression. Furthermore, HDAC6 promoted microvascular endothelial cell proliferation and attenuated the inhibition of proliferation caused by clinorotation unloading. To comprehensively identify microRNAs (miRNAs) that are regulated by HDAC6, we analyzed differential miRNA expression in microvascular endothelial cells after transfection with HDAC6 siRNA and selected miR-155-5p, which was the miRNA with the most significantly increased expression. The ectopic expression of miR-155-5p inhibited microvascular endothelial cell proliferation and directly downregulated Ras homolog enriched in brain (RHEB) expression. Moreover, RHEB expression was downregulated under mechanical unloading and was essential for the miR-155-5p-mediated promotion of microvascular endothelial cell proliferation. Taken together, these results are the first to elucidate the role of HDAC6 in unloading-induced cell growth inhibition through the miR-155-5p/RHEB axis, suggesting that the HDAC6/miR-155-5p/RHEB pathway is a specific target for the preventative treatment of cardiovascular deconditioning.

## 1. Introduction

Mechanical unloading implies decreasing the load on each part of the body, due to microgravity environments or prolonged −6° head-down bed rest, and can induce almost all human physiological system disorders, especially cardiovascular deconditioning [1,2]. Cardiovascular deconditioning is partly secondary to the transfer of fluid from the lower extremities to the cardiothoracic area in the initial period of weightlessness exposure, the most serious symptoms of which are orthostatic intolerance, decreased exercise capacity, and increased resting heart rate, which prevent further interstellar missions and cause a major global public health problem [1,3]. The vascular endothelium, a highly dynamic tissue, is highly sensitive to mechanical forces [4]. Endothelial cells (ECs) form the inner layer of blood vessels, and their impaired function is an important underlying mechanism of cardiovascular disease [5,6]. Mechanical unloading means decreased physical activity with reduced hemodynamic activity and local vascular shear stress, which can result in changes in the morphology and ultrastructure of vascular endothelial cells, as well as the functions of secretion, proliferation, apoptosis and angiogenesis [2,7]. Various studies have claimed that weightlessness can impair the microcirculatory function. Since the microvascular endothelium plays a vital role in the regulation of vascular homeostasis and local blood flow, microvascular endothelial dysfunction is related to cardiovascular disorders [4,8]. Previous studies have shown that unloading conditions inhibit the proliferation of microvascular endothelial cells and promote apoptosis associated with the impairment of endothelial function and might contribute to the cardiovascular deconditioning induced by weightlessness [4,9].

However, the mechanisms of how weightlessness affects the function of the microvascular endothelium remain unclear. Considering that bEnd.3 cell is an internationally recognized immortalized and stable microvascular endothelial cell line with various characteristics of microvascular endothelial cells, it is an ideal model for studying the functions of microvascular endothelial cells in vitro [10,11]. To elucidate the effect of simulated microgravity on brain microvascular endothelial cells, we examined the changes in microvascular endothelial cell function and explored the underlying mechanisms using 2D clinorotation, which was considered a useful device for simulating weightlessness. Given the substantial social and economic burdens associated with cardiovascular deconditioning, there is an urgent need to explore the underlying molecular mechanisms to provide a scientific basis for developing more effective measures.

Chromatin remodeling plays an important role in transcriptional regulation, mainly through the post-translational modification of histones [12]. Several studies have shown that simulated microgravity may induce epigenetic changes, such as DNA methylation and histone acetylation, but the underlying mechanisms remain unclear [13,14]. Histone deacetylases (HDACs) regulate a variety of biological processes by inhibiting or disinhibiting the expression of coding and noncoding genes [15]. HDACs are categorized into four classes according to their homology with yeast histone deacetylases [12,16]. Histone deacetylase 6 (HDAC6) belongs to class IIB, which is cytoplasmic and nuclear enzymes, based on its unique structural and physiological functions [17,18]. HDAC6 interacts with histones as well as nonhistones, and is involved in various biological mechanisms, including cell proliferation, migration, apoptosis, transcription, the degradation of misfolded protein aggregates and the cellular oxidative stress pathway [19]. To our knowledge, HDAC6 plays a crucial role in angiogenesis by regulating EC migration and sprouting [20]. In addition to the well-known regulation of EC migration, HDAC6 has recently been reported to regulate EC proliferation. HDAC6 recruitment enhances EC injury and atherosclerosis by inhibiting vascular endothelial growth factor (VEGF) gene transcription in human umbilical vein endothelial cells (HUVECs) [21]. Furthermore, silencing HDAC6 inhibited proliferation and vasoformation of HUVECs, indicating that HDAC6 is involved in angiogenesis through EC proliferation [22]. However, it is still unclear whether the expression of HDAC6 changes under mechanical unloading and what the role of HDAC6 in microvascular endothelial cells is. Recent studies have shown that HDAC6 serves as a crucial regulator in chromatin maintenance and function by regulating histone acetylation [21,23]. In addition to directly regulating mRNA gene expression, the inhibition of HDAC6 activity has been shown to regulate microRNA (miRNA) expression in a variety of human diseases. HDAC6 negatively regulated miR-27b expression through Rel A/p65 and regulated the MET/PI3K/AKT pathway in diffuse large B-cell lymphoma [24]. The loss or inhibition of HDAC6 expression resulted in increased expression of let-7i-5p, which directly inhibited the tumor suppressor thrombospondin-1 (TSP1), suggesting that the HDAC6/let-7i-5p/TSP1 pathway mediated antitumor and phagocytic activities in patients with hepatocellular carcinoma [25]. The activation of transcription factor 3, which is associated with HDAC6, can reduce the transcription of miR-199a-2/miR-214 [18]. Thus, we hypothesized that HDAC6 was a potent histone modifier whose loss or inhibition led to the reactivation of many functional miRNAs, especially mechanoresponsive microRNAs, and therefore regulated the function of microvascular endothelial cells under mechanical unloading. This provides an insight as to whether HDAC6 acts as an effective histone modifier to regulate functional miRNAs, especially mechanoresponsive microRNAs, thereby regulating the function of microvascular endothelial cells under mechanical unloading.

MicroRNAs (miRNAs) are a class of small noncoding RNAs that are 21–23 nucleotides in length and that induce transcriptional repression and mRNA degradation by binding to the 3′-untranslated regions (UTRs) of target miRNAs [26]. Accumulating evidence has demonstrated that miRNAs play important roles in multiple biological processes related to vascular biological functions, such as angiogenesis, endothelial cell function, and vascular inflammation [27,28]. Additionally, some miRNAs, such as miR-503-5p, miR-27b-5p, miR-145-5p, and miR-151a, are sensitive to mechanical unloading and exert significant effects on endothelial cells [29,30,31]. miR-155-5p is located in the integrated cluster region of B cells on chromosome 21 and is involved in tumorigenesis [32]. Studies have shown that miR-155-5p controls cell proliferation, apoptosis and autophagy in a variety of diseases [33,34,35]. However, the effects of mechanical unloading on the expression of miR-155-5p in microvascular endothelial cells have not been reported.

Ras homolog enriched in brain (RHEB) is commonly known as a member of the Ras family of small GTP binding proteins and includes RHEB1 and RHEB2 [36]. RHEB is involved in cell proliferation, differentiation, and apoptosis, which can be activated by growth factors, such as epithelial growth factor, fibroblast growth factor and brain-derived neurotrophic factor [37,38]. RHEB is dysregulated in various diseases, including tumors, metabolic diseases, and several neurodegenerative diseases [39]. Furthermore, a previous study demonstrated that RHEB acts as the upstream activator of mammalian target of rapamycin complex 1 and is involved in regulating cell growth and apoptosis via the RHEB-mTORC1 signaling pathway [36,40]. RHEB directly mediates the expression of mTOR, and overexpression of RHEB can save inactivated mTOR that could directly regulate cell proliferation, growth and survival [41,42]. However, whether mechanical unloading affects the expression of RHEB and the effect of RHEB on the proliferation of microvascular endothelial cells remains unclear.

In this study, we first demonstrated that HDAC6 expression was downregulated and promoted microvascular endothelial cell proliferation under conditions of mechanical unloading, suggesting that HDAC6 might be a promising therapeutic target for the prevention of unloading-induced cardiovascular deconditioning. Deep RNA sequencing results obtained after HDAC6 knockdown revealed that 11 miRNAs might be regulated by HDAC6 in microvascular endothelial cells. Through analysis of the available literature and unloading experiments, it was found that miR-155-5p was the miRNA with the most significantly regulated expression. Our study further indicated that miR-155-5p was a mechanosensitive miRNA that could inhibit the proliferation of microvascular endothelial cells by directly targeting RHEB expression, highlighting the potential therapeutic value of the HDAC6/miR-155-5p/RHEB signaling pathway in the pathophysiological process of unloading-induced cardiovascular deconditioning.

## 2. Results

### 2.1. Mechanical Unloading Inhibits Microvascular Endothelial Cell Proliferationa and Downregulates HDAC6 Expression

To investigate whether mechanical unloading inhibited proliferation in microvascular endothelial cells, the expression of proliferating cell nuclear antigen (PCNA) was examined by Western blotting after clinorotation for 48 h, and the results showed that PCNA expression was significantly downregulated during unloading (Figure 1A and Appendix A). Subsequently, the Cell Counting Kit-8 (CCK-8) assay was used to detect the effects of mechanical unloading on microvascular endothelial cell proliferation. The growth of cell continuously increased in a time-dependent manner, while that in the con group increased more obviously, indicating that mechanical unloading inhibited microvascular endothelial cell proliferation (Figure 1B). Correspondingly, the 5-Ethynyl-2′-deoxyuridine (EdU) labeling assay results were consistent with the CCK-8 assay and PCNA protein expression results. The number of EdU-positive cells under unloading conditions was decreased compared with that under control conditions (Figure 1C).

As HDACs are potent angiogenesis and cell proliferation modifiers, more angiogenesis signaling pathways might be governed by HDACs during unloading. However, the specific effect of HDACs in microvascular endothelial cells remains unclear. qRT-PCR showed that HDAC6 expression was decreased from 24 to 72 h and reached its lowest level at 48 h under mechanical unloading conditions (Figure 1D). Furthermore, the protein expression of HDAC6 was markedly decreased compared with that in the control group (Figure 1E and Appendix A). Together, these findings indicate that HDAC6 may be involved in regulating microvascular endothelial cell proliferation under mechanical unloading conditions.

### 2.2. HDAC6 Promotes Microvascular Endothelial Cell Proliferation and Attenuates the Inhibition of Cell Proliferation Caused by Clinorotation Unloading

To examine the role of HDAC6 in microvascular endothelial cell proliferation, we first examined the localization of the HDAC6 protein via immunofluorescence staining and Western blotting. The results showed that HDAC6 localized to the nucleus and cytoplasm to regulate many important biological processes (Figure 2A,B). Microvascular endothelial cells were transfected with HDAC6 siRNA or pEX-HDAC6, and the results showed that compared with the negative control, siRNA-HDAC6 significantly decreased the protein expression of PCNA, while pEX-HDAC6 increased its expression (Figure 2C and Appendix A). The CCK-8 assay demonstrated that the knockdown of HDAC6 inhibited the growth of bEnd.3 cells, whereas the overexpression of HDAC6 significantly enhanced the growth of cells (Figure 2E). The results of the EdU labeling assays also showed similar trends (Figure 2D).

Furthermore, to explore whether the overexpression of HDAC6 could rescue the inhibited cell proliferation caused by unloading in vitro, bEnd.3 cells were transfected with pEX-HDAC6 and then cultured under mechanical unloading conditions for 48 h. HDAC6 overexpression substantially reversed the reduction in the protein expression of PCNA that was induced by clinorotation-unloading (Figure 2F and Appendix A). Similar results were obtained from the CCK-8 and EdU labeling assays (Figure 2G,H). Taken together, these results indicated that as a mechanoresponsive molecule, HDAC6 promoted microvascular endothelial cell proliferation, while silencing HDAC6 expression inhibited proliferation in vitro.

### 2.3. HDAC6 Inhibition Increases Intranuclear Histone Expression and Increases miR-155-5p Expression in Microvascular Endothelial Cells

Since HDAC6 played an important epigenetic role in regulating the transcription of many genes, we explored the expression of intranuclear histones by Western blotting analysis and found that silencing HDAC6 expression increased the protein expression levels of Ac-H3K9, while upregulating HDAC6 expression decreased the protein expression levels of Ac-H3K9 (Figure 3A and Appendix A). Thus, to comprehensively identify miRNAs regulated by HDAC6, siRNA-HDAC6 and the corresponding control were transfected into bEnd.3 cells, and miRNA microarrays were performed. A total of 11 miRNAs were significantly upregulated in siRNA-HDAC6-transfected bEnd.3 cells (*p <* 0.05) (Figure 3B,C). Then, we analyzed whether these miRNAs were related to cell proliferation according to the previously published literature, verified whether their expression was upregulated after siRNA-HDAC6 transfection and determined whether their expression changed under clinorotation unloading (Figure 3D,E). According to this analysis, five miRNAs were identified as miRNAs potentially regulated by HDAC6 under mechanical unloading conditions. Considering the changes after both HDAC6 knockdown and weightlessness, we chose miR-155-5p, which is the miRNA whose expression was most significantly increased, as our candidate miRNA.

Next, to further investigate whether miR-155-5p expression is directly regulated by HDAC6 in bEnd.3 cells, we performed a chromatin immunoprecipitation (ChIP) assay with an HDAC6 antibody. Treatment with anti-HDAC6 showed significantly enriched binding of HDAC6 to the miR-155-5p promoter regions compared with the control (Figure 3F,G). It is well-known that HDAC6 could deacetylate H3K9, and H3K9 deacetylation at the promoter usually results in reduced transcription [21,25,43]. Therefore, we then performed ChIP assays with Ac-H3K9 in the context of HDAC6 knockdown in bEnd.3 cells. Results indicated that Ac-H3K9 was enriched in the sites of the miR-155-5p promoter. Moreover, knockdown of HDAC6 significantly increased H3K9 acetylation in the miR-155-5p promoter regions in bEnd.3 cells. Knockdown of HDAC6 reduced histone deacetylations in the miR-155-5p promoter regions in bEnd.3 cells (Figure 3H).

### 2.4. miR-155-5p Inhibits Microvascular Endothelial Cell Proliferation

miRNA-155-5p is predominantly described as an oncogenic or tumor suppressor miRNA in several cancers. However, the exact role of miR-155-5p in the proliferation of microvascular endothelial cells has not been elucidated. To validate the aberrant expression of miR-155-5p under clinorotation unloading conditions, we performed quantitative RT-PCR analysis of the expression of miR-155-5p from 24 to 72 h, and the results showed a significant difference between the control and clinorotation groups, especially at 48 h (Figure 4A). To further examine the role of miR-155-5p in microvascular endothelial cells, we transfected an miR-155-5p mimic or inhibitor into bEnd.3 cells. The results illustrated that compared with the NC-mimic group, the miR-155-5p mimic decreased the protein expression of PCNA and the number of EdU-positive cells under unloading conditions, whereas the miR-155-5p inhibitor showed the opposite effect (Figure 4B,D and Appendix A). A similar result was obtained in the CCK-8 assay (Figure 4C). Taken together, these results show that miR-155-5p played a negative regulatory role in microvascular endothelial cell proliferation.

### 2.5. miR-155-5p Directly Downregulates RHEB Expression in Microvascular Endothelial Cells

To identify the molecular mechanisms underlying miR-155-5p-mediated bEnd.3 cell proliferation, we used the target prediction program TargetScan and the miRDB database to predict the mRNA targets of miR-155-5p. We then selected molecules involved in the cellular proliferation process through literature reviews and unloading experiments. RHEB, a critical regulator of cell proliferation, was identified as one of the potential targets of miR-155-5p. After transfection with an miR-155-5p mimic or inhibitor, we used real-time PCR and Western blotting to detect the mRNA and protein expression levels of RHEB. Consequently, miR-155-5p considerably influenced RHEB mRNA and protein expression compared to the NC group (Figure 5A,B and Appendix A). To determine whether miR-155-5p directly targets RHEB, the RHEB 3′-UTR-containing wild-type or mutant miR-155-5p-binding sites were cloned into a luciferase reporter vector. The relative luciferase activity of the reporter containing the wild-type 3′-UTR of RHEB was reduced compared with that of the control group in 293T cells (Figure 5C,D). The results indicated that RHEB was a vital target of miR-155-5p for sensing mechanical unloading and regulating microvascular endothelial cell proliferation.

### 2.6. RHEB Is Downregulated under Mechanical Unloading Conditions and Essential for the miR-155-5p-Mediated Inhibition of Microvascular Endothelial Cell Proliferation

During mechanical unloading, the expression of miR-155-5p was continuously increased, while the mRNA level of RHEB was continuously decreased (Figure 6A). Moreover, the protein expression of RHEB was significantly decreased in the clinorotation group (Figure 6B and Appendix A). To further examine the role of RHEB, we used RNA interference to knockdown RHEB in bEnd.3 cells. The results confirmed that siRNA-RHEB transfection decreased the expression level of PCNA protein (Figure 6C and Appendix A). Consistently, CCK-8 and EdU labeling assays demonstrated that the suppression of RHEB expression inhibited the growth of bEnd.3 cells (Figure 6D,E). To confirm that the suppression of proliferation by miR-155-5p depends on RHEB in microvascular endothelial cells, inhibitor-155-5p and siRNA-RHEB or its negative control were co-transfected into bEnd.3 cells. SiRNA-RHEB markedly attenuated the inhibitor-155-5p-induced increase in the PCNA protein levels. Furthermore, CCK-8 and EdU labeling assays showed similar changes (Figure 6F–H and Appendix A).

### 2.7. HDAC6 Promotes Microvascular Endothelial Cell Proliferation by Inhibiting the mir-155-5p/RHEB Axis

Since HDAC6 interacted with miR-155-5p and RHEB was the target of miR-155-5p, we explored whether HDAC6 could positively regulate the expression of RHEB. Our results demonstrated that HDAC6 partially reversed the reduction in RHEB protein expression induced by clinorotation unloading in bEnd.3 cells through miR-155-5p (Figure 7A and Appendix A). In addition, to further verify whether HDAC6 could promote proliferation by inhibiting the miR-155-5p/RHEB axis in microvascular endothelial cells, we co-transfected siRNA-HDAC6, inhibitor-155-5p, and their negative controls into bEnd.3 cells. qRT-PCR and Western blotting analyses showed that inhibitor-155-5p rescued the siRNA-HDAC6-induced decrease in RHEB mRNA and protein expression (Figure 7B,C and Appendix A). Moreover, inhibitor-155-5p reversed the proliferation-suppressing effects of siRNA-HDAC6, as evidenced by PCNA protein expression, CCK-8 and EdU labeling analyses (Figure 7C–E). In addition, the enhanced microvascular endothelial cell proliferation induced by pEX-HDAC6 during unloading was reversed by siRNA-RHEB, including the protein level of PCNA and CCK-8 and EdU labeling assays (Figure 7F–H and Appendix A). In conclusion, these data demonstrated that RHEB was responsible for HDAC6-mediated microvascular endothelial cell proliferation during unloading.

## 3. Discussion

As a highly dynamic tissue, the vascular endothelium is highly sensitive to mechanical forces, and obvious morphological and functional changes occur during mechanical conduction, resulting in cardiovascular dysfunction [2]. Previous studies have shown that mechanical unloading can elicit macrovascular endothelial cell proliferation [5,44]. However, there is little information that addresses the effect of unloading on microvascular endothelial cells. In the current study, we observed that mechanical unloading inhibited microvascular endothelial cell proliferation and downregulated HDAC6 expression, and we confirmed its direct impacts on microvascular endothelial cell proliferation during unloading in vitro. Mechanistic studies showed that HDAC6 inhibited miR-155-5p expression to promote cell proliferation by upregulating the mRNA and protein expression of RHEB. Thus, our study was the first to elucidate the role of the HDAC6/miR-155-5p/RHEB pathway in the treatment of cardiovascular deconditioning (Figure 8).

HDACs regulate the post-translational modification of lysine residues in histone tails through deacetylation, thereby regulating gene expression. Accumulating evidence shows that HDAC6 is involved in the prognosis and progression of a variety of diseases through various regulatory mechanisms, such as cell transcription, proliferation, migration, apoptosis, cellular oxidative stress and degradation of misfolded proteins through aggregation, and may serve as a target for future gene therapy [18,45]. Moreover, HDAC6 can interact with several nonhistone proteins, such as heat shock protein 90 (HSP90), extracellular signal-related kinase 1 (ERK1), Cortactin, α-tubulin, Peroxiredoxins, Survivin, heat shock transcription factor-1 (HSF-1), KRAS, and Miro-1 [17]. Although HDAC6 is a major cytoplasmic enzyme, it is also known for its transcriptional activity in the nucleus. Recent studies have shown that HDAC6 directly controls the activity of transcription inhibitors by interacting with various corepressors, such as ligand-dependent nuclear receptor corepressor (LCoR), runt-related transcription factor 2 (Runx2), nuclear factor kappa-light-chain-enhancer of activated B cells (NF-κB), and G3BP1 [46,47,48,49]. Gain- and loss-of-function experiments using HDAC6 siRNA or pEX-HDAC6, respectively, confirmed that HDAC6 promoted the proliferation of microvascular endothelial cells. Previous studies reported that HDAC6 negatively modulated the expression of some miRNAs to regulate pathological states. Therefore, we hypothesized that the loss or inhibition of HDAC6 could reactivate mechanosensitive miRNAs, thereby regulating gene expression that contributed to microvascular endothelial cell growth during mechanical unloading. Among the candidate miRNAs for HDAC6-specific regulation, we selected miR-155-5p as it could be inhibited again by the restoration of HDAC6 expression in microvascular endothelial cells and was overexpressed during rotation. Similarly, we further demonstrated that HDAC6 directly regulated and inhibited miR-155-5p expression in microvascular endothelial cells via the ChIP assay.

miR-155-5p is mainly considered an oncogenic miRNA in several cancers, such as non-small cell lung cancer, liver cancer, colorectal cancer, oral squamous cell carcinoma and breast cancer [50,51,52]. It has also been reported that the overexpression of miR-155-5p can enhance the proliferation, migration and ECM secretion of osteoarthritis chondrocytes and inhibit cell apoptosis [53]. In addition, miR-155-5p exerts tumor-suppressor effects in gastric cancer [54,55]. miR-155-5p inhibition could revitalize senescent mesenchymal stem cells and enhance cardiac protection after infarction [56]. Furthermore, miR-155-5p inhibited the viability of vascular smooth muscle cells and promoted the formation of aneurysms by targeting Fos and ZIC3 [27]. Consequently, miR-155-5p may perform different functions and possess different activities depending on the cell type. However, the effects of miR-155-5p on microvascular endothelial cell proliferation, especially under unloading conditions, have not been previously reported. Our investigations suggested that miR-155-5p expression was sensitive to unloading conditions and that HDAC6 could decrease the expression of miR-155-5p during unloading. Furthermore, we demonstrated that miR-155-5p could impair the functions of microvascular endothelial cells by targeting RHEB.

RHEB is a molecular switch that regulates cell proliferation, differentiation, and apoptosis. The depletion of RHEB in combination with rapamycin treatment not only led to reduced astrocyte proliferation but also to reduced neuronal protection [57]. Furthermore, RHEB can effectively promote the survival of colon cancer cells via an mTORC1–independent mechanism [42]. Sufficient evidence has confirmed that the mTOR signaling pathway plays an important role in cell proliferation. mTOR is a serine/threonine protein kinase that regulates cell processes, such as cell proliferation, growth and survival [58,59]. It is most often activated in human cancers and has been proven to be an anticancer therapeutic target [60,61,62]. Here, we reported that both the protein and mRNA levels of RHEB were reduced under unloading, and RHEB was a direct target molecule of miR-155-5p in vitro. miR-155-5p overexpression inhibited the mRNA and protein expression of RHEB, while the miR-155-5p inhibitor promoted the expression of RHEB in microvascular endothelial cells. Our study further demonstrated that siRNA RHEB could inhibit proliferation.

## 4. Materials and Methods

### 4.1. Cell Culture

The mouse brain microvascular endothelial bEnd.3 cell line was acquired from the Cell Bank of the Chinese Academy of Sciences (Shanghai, China). The cells were cultured in high-glucose DMEM with 10% FBS (Gibco, Thermo Fisher, Waltham, MA, USA) and 100 units/mL penicillin/streptomycin (Gibco, Thermo Fisher, Waltham, MA, USA) in a 95% humidified incubator with 5% CO_2_ at 37 °C. Cells at passages 4–8 were used in the experiments, and all the experiments were repeated 3 times (N = 3).

### 4.2. D Clinorotation

The process of 2D clinorotation is widely used to simulate an unloading environment for cells on the ground. bEnd.3 cells were seeded at a density of 1 × 10^5^ cells on special rotating coverslips, which were placed in 6-well plates and cultured routinely. After the cells were completely attached to the coverslips and grown to approximately 50% confluence, the coverslips were inserted into the scaffold of the rotating chambers (Astronaut Research and Training Center, Beijing, China), which were completely filled with DMEM supplemented with 10% FBS, and the lids of vessels were tightened after the air bubbles were completely removed. Finally, the chambers were placed into the 2D clinorotation, which was placed in an incubator at 37 °C and rotated around a horizontal axis at 24 rpm for 24, 48, and 72 h. The control group was placed in the same incubator without rotation.

### 4.3. Quantitative Real-Time PCR (qRT-PCR) Analysis

Total RNA was isolated from bEnd.3 cells with RNAiso Plus (Takara, Tokyo, Japan) and reverse transcribed according to the standard protocol of the manufacturer. The PrimeScript™ RT Master Mix Kit (Takara, Tokyo, Japan) was used to reverse transcribe mRNA to cDNA under the following conditions: 37 °C for 15 min, 85 °C for 5 s, and holding at 4 °C. miRNA was reverse-transcribed using a Mir-X miRNA First-Strand Synthesis Kit (Takara, Tokyo, Japan) under the following conditions: 37 °C for 1 h, 85 °C for 5 min, and holding at 4 °C. The mRNA and miRNA expression levels were detected by a CFX96 real-time PCR detection system (Bio-Rad Laboratories, Hercules, CA, USA) and SYBR^®^ Premix Ex Taq TM II (Takara, Tokyo, Japan). The relative Ct (2^−ΔΔCt^) method was used to calculate the relative expression level of miRNAs or mRNAs between samples, and GAPDH or U6 small nuclear RNA expression was used as endogenous controls. The primer sequences used in this study are shown in Appendix A.

### 4.4. Western Blotting Analysis

bEnd.3 cells were lysed using M-PER Mammalian Protein Extraction Reagent (Thermo Fisher Scientific, Waltham, MA, USA) containing a 10% protease inhibitor cocktail (Roche, Mannheim, Germany) to extract the total proteins, and a Pierce™ BCA Protein Assay Kit (Thermo Fisher Scientific, Waltham, MA, USA) was used to quantify the concentrations of the proteins according to the standard instructions. Equal amounts of protein samples were separated by NuPage Bis-Tris polyacrylamide gels (Invitrogen, Carlsbad, CA, USA) and transferred onto polyvinylidene fluoride (PVDF) membranes. The blots were blocked with 5% skim milk for 2 h at room temperature and incubated with the following primary antibodies at 4 °C overnight: GAPDH (1:1000; Cell Signaling Technology, Boston, MA, USA), HDAC6 (1:1000; Cell Signaling Technology, Boston, MA, USA), RHEB (1:1000; Cell Signaling Technology, Boston, MA, USA), Ac-H3K9 (1:1000; Cell Signaling Technology, Boston, MA, USA), PCNA (1:1000; Cell Signaling Technology, Boston, MA, USA), and Lamin B (1:10000; Proteintech, Chicago, IL, USA). The membranes were then incubated with goat anti-rabbit or anti-mouse horseradish peroxidase-conjugated secondary antibodies (1:5000; ZS-GB-BIO, Beijing, China) at room temperature for 2 h. The signals were visualized by an ECL detection kit (Thermo Fisher Scientific, Waltham, MA, USA), and the intensities of blots were quantified using Image J software (Wikimedia Foundation, San Francisco, CA, USA).

### 4.5. Cell Transfection

pEX-HDAC6, siRNA-H DAC6, siRNA-RHEB, miR-155-5p mimic, miR-155-5p inhibitor, and their corresponding controls were purchased from GenePharma (Shanghai, China). bEnd.3 cells were seeded and cultured in 6-well plates overnight and transiently transfected with pEX-HDAC6 (100 ng/μL), siRNA-HDAC6 (80 nM), siRNA-RHEB (80 nM), miR-155-5p mimic (40 nM), miR-155-5p inhibitor (80 nM) and their corresponding negative controls using the transfection reagent Lipofectamine 2000 (Invitrogen, Carlsbad, CA, USA) following the manufacturer’s protocol. The transfection efficiency was detected by qRT-PCR after 48 h of cell transfection. The sequences of the siRNAs specific for HDAC6, RHEB, and miR-155-5p and the negative controls are listed in Appendix A.

### 4.6. Cell Proliferation Assay

A CCK-8 assay is a typical method for measuring cell viability based on the chromogenic reaction of WST-8, and the assay was performed following the manufacturer’s protocol. Cells were plated on 96-well plates at a density of 2 × 10^4^ cells/mL (100 µL/well) and cultured overnight at 37 °C. After incubation for 24, 48 and 72 h, 10 μL of CCK-8 reagent (Dojindo, Shanghai, China) was added to each well and cultured for an additional 2 h at 37 °C. Cell viability was detected at 450 nm to calculate the optical density (OD) values using a microplate reader. All the groups included three biological replicates, and each experiment was performed in triplicate.

### 4.7. Ethynyl-2′-deoxyuridine (EdU) Labeling

An EdU cell proliferation kit with Alexa Fluor (Beyotime, Shanghai, China) was used to detect cell proliferation; in this assay, the thymidine analog Edu replaced thymine and was incorporated into DNA during the process of DNA synthesis in order to evaluate DNA replication. bEnd.3 cells were plated in 6-well plates at 1 × 10^5^ cells per well and cultured overnight. One milliliter of EdU working solution (50 μM) was added to each well and incubated for another 4 h at 37 °C. Subsequently, the cells were fixed with 4% paraformaldehyde for 15 min, washed with PBS solution containing 3% bovine serum albumin (BSA), and permeabilized in 0.3% Triton X-100 solution for 15 min. EdU nuclear staining was measured according to the standard manufacturer’s protocol. Finally, the images were captured and merged using a confocal microscope. The ratio of the number of EdU-positive cells to the total number of DAPI-positive cells (blue) was used to indicate the EdU incorporation rate. Each experiment was performed in triplicate.

### 4.8. Luciferase Assay

TargetScan and miRDB were used to predict that the 3′-UTR of RHEB contains miR-155-5p binding sites. The 293T cells were co-transfected with miR-155-5p reagents (mimic, inhibitor or their negative controls) and the wild-type (WT) RHEB 3′-UTR or mutant (MUT) RHEB 3′-UTR, which were inserted into the pmirGLO vector using GP-transfect-Mate transfection reagent. After co-transfection and culture for 48 h, the luciferase activity was tested using a dual-luciferase reporter assay system following the manufacturer’s instructions, and the firefly luciferase activity levels were normalized to the co-expressed Renilla luciferase activity in each sample.

### 4.9. Chromatin Immunoprecipitation (ChIP) Assay

ChIP assays were conducted according to the manufacturer’s protocol (SimpleChIP Enzymatic Chromatin IP Kit; Cell Signaling Technology, Boston, MA, USA). bEnd.3 cells were cross-linked with formaldehyde for 10 min and washed with ice-cold PBS. The nuclei were processed, and DNA samples were cleaved to the appropriate length of 100–900 bp by enzymatic hydrolysis. DNA samples were purified, and agarose gel electrophoresis was performed to detect DNA fragment size. Cross-linked DNA was incubated overnight using HDAC6 (anti-HDAC6) antibodies (NOVUS, Denver, CO, USA) and acetylated histone H3K9 (Ac-H3) antibodies (Cell Signaling Technology, Boston, MA, USA), and protein A-agarose beads were added to the reaction mixtures and incubated for another 2 h. After extraction and purification, the DNA was amplified, and relative gene expression was measured by real-time quantitative PCR using primers specific for the binding motif of HDAC6 in the miR-155 promoter region. The primers used for PCR were specific for the miR-155 promoter sequence: forward: 5′-cagggacgcaggtaggc-3′ and reverse: 5′-ctccatcaggactccctctc-3′.

### 4.10. Immunofluorescence Staining

Cells were seeded in 6-well plates at 1 × 10^5^ cells/well. When the cells reached approximately 80% confluence, they were fixed with 4% formaldehyde at room temperature for 15 min and washed three times with PBS. The cells were treated with 0.5% Triton X-100, blocked with 5% goat serum for 30 min, and then incubated with an anti-HDAC6 antibody (1:200, NOVUS, Denver, CO, USA) at 4 °C overnight. The cells were then washed, incubated with a secondary antibody conjugated to FITC (Proteintech, Chicago, IL, USA) and incubated with DAPI (Beyotime, Shanghai, China). Images were captured and analyzed with a fluorescence microscope.

### 4.11. Nuclear-Cytoplasmic Fractionation

A nuclear and cytoplasmic protein extraction kit (Beyotime, Shanghai, China) was used to extract the proteins according to the manufacturer’s instructions. Cells were collected by centrifugation, and cytoplasmic protein extraction reagent A with PMSF was added to the cells. After incubation in an ice bath for 15 min, cytoplasmic protein extraction reagent B was added, and the samples were centrifuged at 12,000× *g* for 5 to obtain the cytoplasmic proteins.

### 4.12. Statistical Analysis

All the experiments were performed in triplicate. The data were analyzed with the GraphPad Prism 8.0 and SPSS 22.0 software and were expressed as the mean standard deviation (SD) from at least three independent experiments. Variance analysis between the control and treated groups was performed with a Student’s *t*-test and one-way ANOVA. *p* < 0.05 and *p* < 0.01 were considered statistically significant.

## 5. Conclusions

In conclusion, this study first reported that HDAC6 expression was downregulated under mechanical unloading and that HDAC6 is a critical regulator of microvascular endothelial cell proliferation by inhibiting miR-155-5p expression to promote the expression of RHEB (Figure 8). Our research revealed the function of the HDAC6/miR-155-5p/RHEB signaling pathway in microvascular endothelial cells and indicated the promising value of HDAC6 in the preventative treatment of cardiovascular deconditioning caused by mechanical unloading.

## Figures and Tables

**Figure 1 ijms-22-10527-f001:**
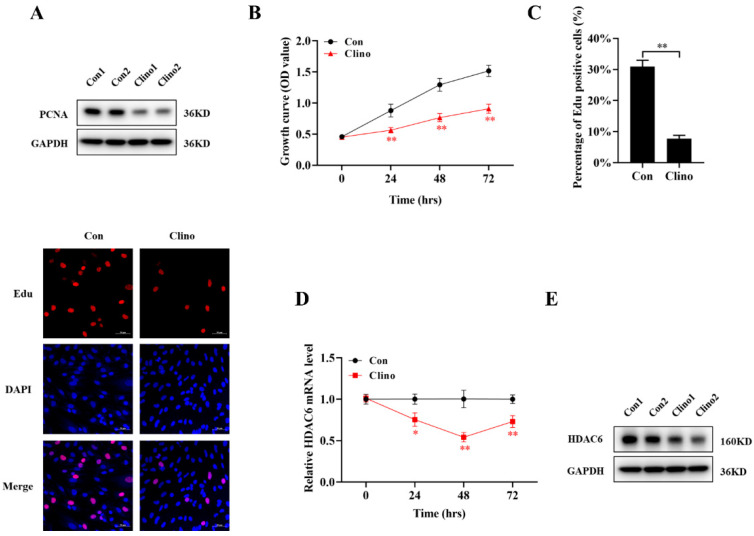
Mechanical unloading inhibits microvascular endothelial cell proliferation and downregulates HDAC6 expression. (**A**) Protein expression of PCNA was analyzed by Western blotting in microvascular endothelial cells that were cultured under clinorotation conditions for 48 h (N = 3). (**B**) Cell proliferation was evaluated by a CCK-8 assay at 24–72 h (N = 3). (**C**) The EdU incorporation assay was analyzed by confocal microscopy. Proliferating microvascular endothelial cells were labeled with EdU. Microvascular endothelial cells were stained with the nucleic acid dyes Hoechst (blue) and EdU (red) (N = 3). Scale bar, 50 µm. (**D**) HDAC6 mRNA expression was assessed in microvascular endothelial cells under clinorotation conditions for 24, 48 and 72 h (N = 3). (**E**) The HDAC6 protein level was analyzed by Western blotting (N = 3). * *p <* 0.05, ** *p <* 0.01 vs. control.

**Figure 2 ijms-22-10527-f002:**
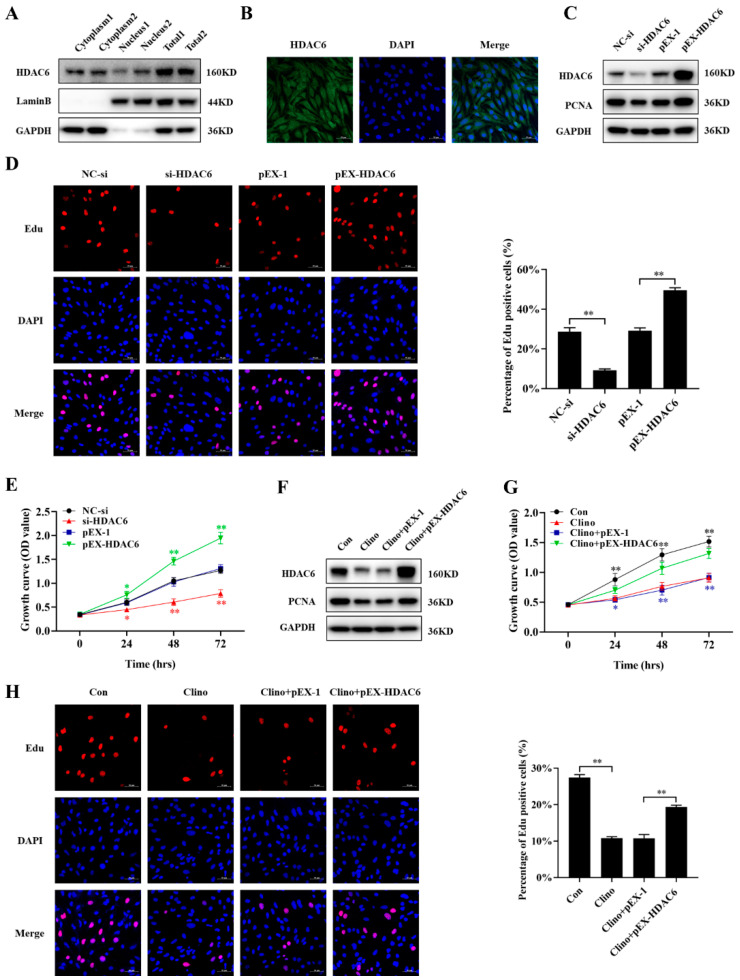
HDAC6 promotes microvascular endothelial cell proliferation and attenuates the inhibition of cell proliferation caused by clinorotation unloading. (**A**) The localization of HDAC6 was examined by Western blotting. (**B**) The localization of HDAC6 was examined by immunofluorescence staining. (**C**) The HDAC6 and PCNA protein levels in microvascular endothelial cells were analyzed by Western blotting after transfection with HDAC6 siRNA or pEX-HDAC6 (N = 3). (**D**) The EdU incorporation assay was analyzed by confocal microscopy (N = 3). (**E**) Cell proliferation was evaluated by CCK-8 assay (N = 3). The symbol *(Red) represents the NC-si vs. si-HDAC6 group; the symbol *(Green) represents the pEX vs. pEX-HDAC6 group. (**F**) The protein expression of HDAC6 and PCNA was analyzed by Western blotting after microvascular endothelial cells were transfected with pEX-HDAC6 and then cultured under mechanical unloading conditions for 48 h (N = 3). (**G**) Cell proliferation was evaluated by CCK-8 assay (N = 3). The symbol *(Black) represents the Con vs. Clino group; the symbol *(Blue) represents the Clino plus pEX vs. Clino plus pEX-HDAC6 group. (**H**) The EdU incorporation assay was analyzed by confocal microscopy (N = 3). Scale bar, 50 µm. * *p <* 0.05, ** *p <* 0.01 vs. control.

**Figure 3 ijms-22-10527-f003:**
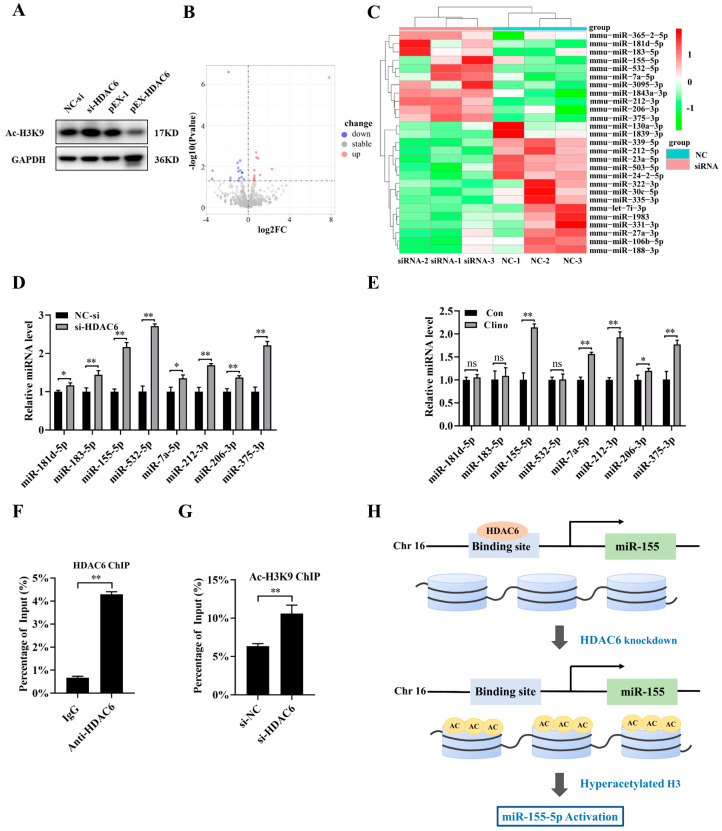
HDAC6 inhibition increases intranuclear histone expression and increases in miR-155-5p expression in microvascular endothelial cells. (**A**) Ac-H3K9 protein expression was analyzed by Western blotting in microvascular endothelial cells treated with HDAC6 siRNA, pEX-HDAC6 or the corresponding controls for 48 h (N = 3). (**B**) Representative volcano plot of miRNA expression levels after HDAC6 siRNA-transfected microvascular endothelial cells based on the RNA-seq results. (**C**) Representative heatmap of miRNA expression levels after HDAC6 siRNA transfection of microvascular endothelial cells based on the RNA-seq results (N = 3). (**D**) Differential miRNA expression analysis of HDAC6 siRNA-transfected microvascular endothelial cells was verified by qRT-PCR (N = 3). (**E**) The differential expression of miRNAs selected from microarray assay data was determined by qRT-PCR in microvascular endothelial cells under clinorotation-unloading conditions for 48 h (N = 3). (**F**) ChIP-qRT-PCR assay to assess HDAC6 association with the miR-155-5p promoter (N = 3). (**G**) ChIP-qRT-PCR assay to assess the enrichment of histone deacetylation by HDAC6 in the miR-155-5p promoter region (N = 3). (**H**) Schematic diagram of HDAC6, the mechanism which regulates miR-155-5p expression through the deacetylation of histones in the miR-155 promoter. * *p <* 0.05, ** *p <* 0.01 vs. control.

**Figure 4 ijms-22-10527-f004:**
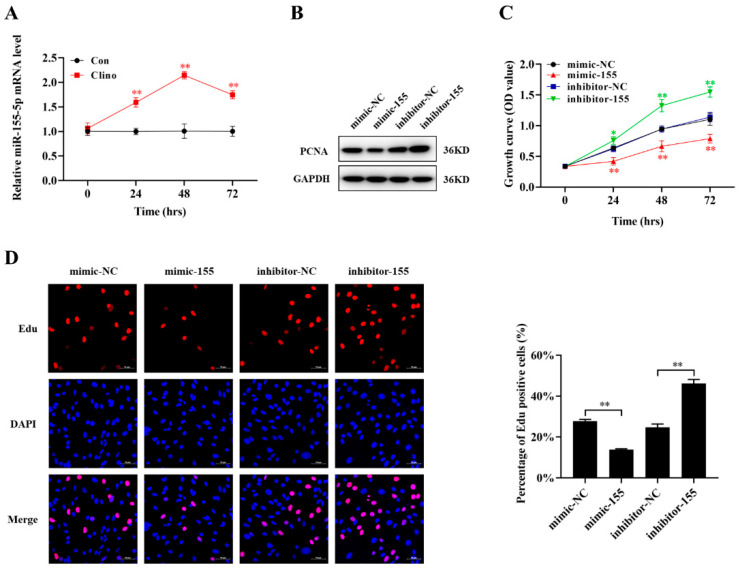
The miR-155-5p inhibits microvascular endothelial cell proliferation. (**A**) miR-155-5p mRNA level was analyzed by qRT-PCR in microvascular endothelial cells that were cultured under clinorotation conditions at 24–72 h (N = 3). (**B**) The protein expression of PCNA in microvascular endothelial cells was analyzed by Western blotting (N = 3). (**C**) Cell proliferation was evaluated by CCK-8 assay (N = 3). The symbol *(Red) represents the mimic-NC vs. mimic-155 group; the symbol *(Green) represents the inhibitor-NC vs. inhibitor-155 group. (**D**) Proliferating microvascular endothelial cells were labeled with EdU (N = 3). Scale bar, 50 µm. * *p <* 0.05, ** *p <* 0.01 vs. control.

**Figure 5 ijms-22-10527-f005:**
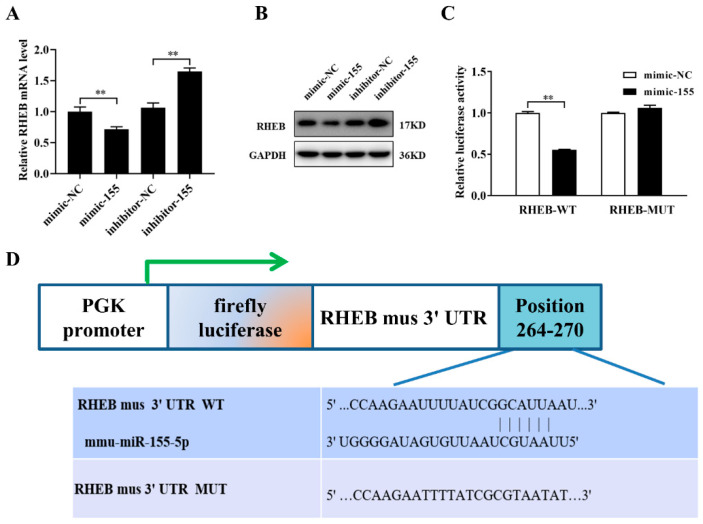
MiR-155-5p directly downregulates RHEB in microvascular endothelial cells. (**A**) qRT-PCR analysis of RHEB mRNA expression in microvascular endothelial cells treated with mimic-155-5p, inhibitor-155-5p or the corresponding controls for 48 h (N = 3). (**B**) Western blotting analysis of the protein expression of RHEB (N = 3). (**C**) After 293T cells were treated with mimic-155-5p and the corresponding controls for 48 h, the luciferase activities of the RHEB WT and MUT reporters were evaluated. (N = 3). (**D**) Schematic representation of the luciferase reporters containing RHEB 3′-UTR WT or MUT sequences. * *p <* 0.05, ** *p <* 0.01 vs. control.

**Figure 6 ijms-22-10527-f006:**
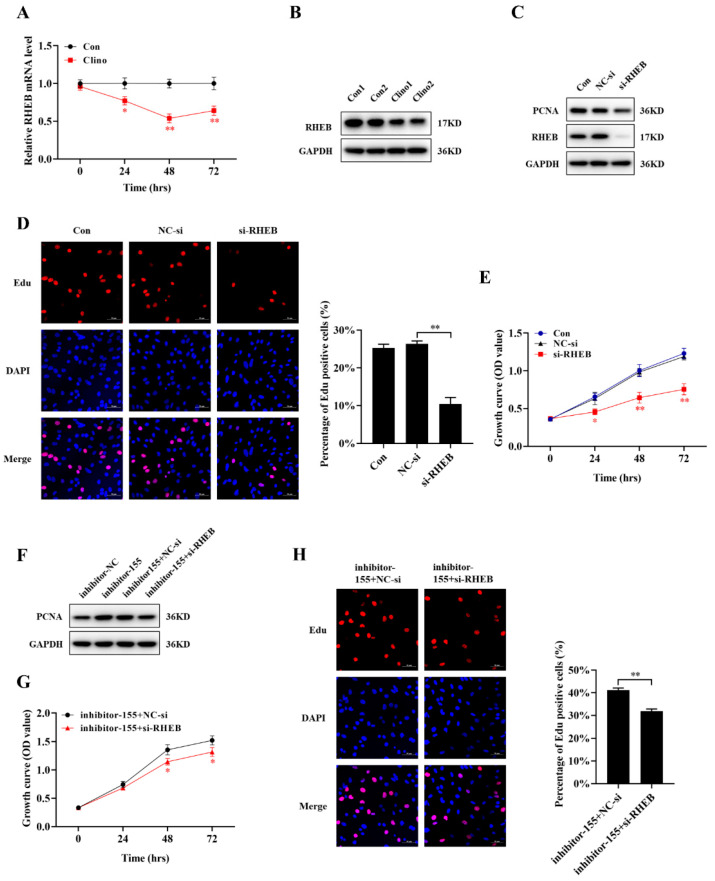
RHEB is downregulated under mechanical unloading and is essential for the miR-155-5p-mediated inhibition of microvascular endothelial cell proliferation. (**A**) RHEB mRNA expression was assessed in microvascular endothelial cells under clinorotation conditions for 24, 48 and 72 h (N = 3). (**B**) The RHEB protein level was analyzed by Western blotting (N = 3). (**C**) The RHEB and PCNA protein levels in microvascular endothelial cells were analyzed by Western blotting after transfection with RHEB siRNA (N = 3). (**D**) The EdU incorporation assay was analyzed by confocal microscopy (N = 3). (**E**) Cell proliferation was evaluated by CCK-8 assay (N = 3). The symbol *(Red) represents the NC-si group vs. si-RHEB group. (**F**) PCNA protein levels were measured by Western blotting after co-transfecting inhibitor-155-5p and siRNA-RHEB into microvascular endothelial cells (N = 3). (**G**) Cell proliferation was evaluated by CCK-8 assay at 24–72 h (N = 3). (**H**) The EdU incorporation assay was analyzed by confocal microscopy (N = 3). Scale bar, 50 µm. * *p <* 0.05, ** *p <* 0.01 vs. control.

**Figure 7 ijms-22-10527-f007:**
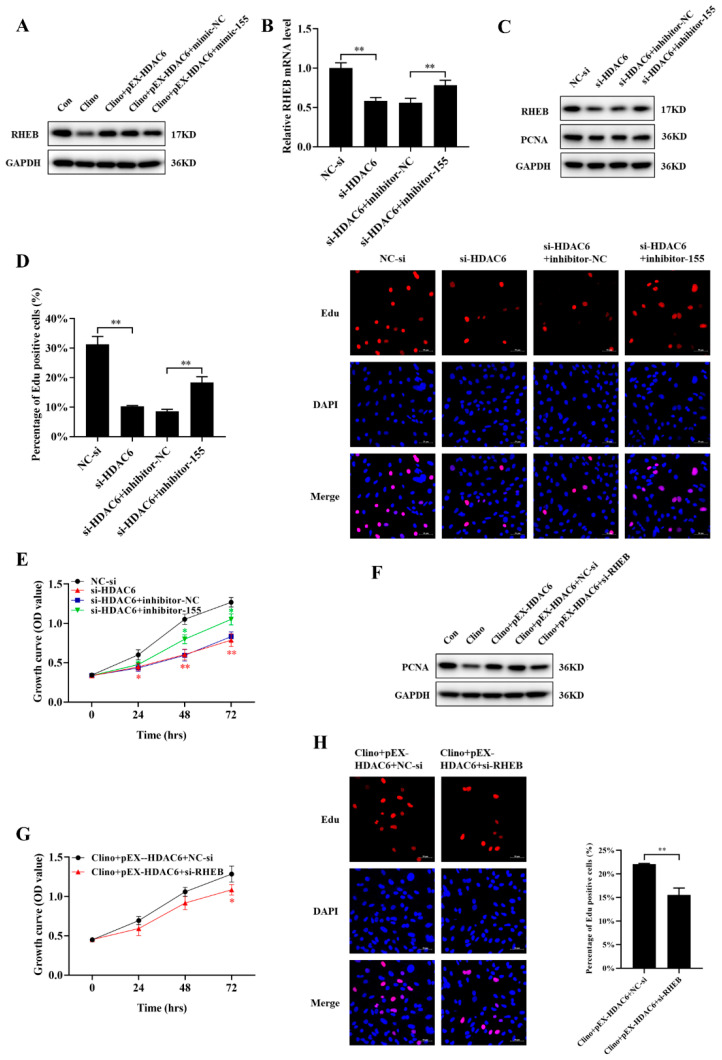
HDAC6 promotes microvascular endothelial cell proliferation by inhibiting the miR-155-5p/RHEB axis. (**A**) RHEB protein expression was analyzed by Western blotting after microvascular endothelial cells were transfected with pEX-HDAC6 alone or pEX-HDAC6 plus mimic-155 and subjected to clinorotation unloading for 48 h (N = 3). **(B**) qRT-PCR analysis of RHEB level after cotransfection with siRNA-HDAC6, inhibitor-155-5p, and their negative control into bEnd.3 cells (N = 3). (**C**) Western blotting analysis of RHEB and PCNA protein expression (N = 3). (**D**) The EdU incorporation assay was analyzed by confocal microscopy (N = 3). Scale bar, 50 µm. (**E**) Cell proliferation was evaluated by CCK-8 assay at 24–72 h (N = 3). The symbol *(Red) represents the NC-si vs. si-HDAC6 group; the symbol *(Green) represents the si-HDAC6 plus inhibitor-NC vs. si-HDAC6 plus inhibitor-155 group. (**F**) PCNA protein expression was analyzed by Western blotting after microvascular endothelial cells were transfected with pEX-HDAC6 alone or pEX-HDAC6 plus siRNA-RHEB and subjected to clinorotation unloading for 48 h (N = 3). (**G**) Cell proliferation was evaluated by CCK-8 assay at 24–72 h (N = 3). (**H**) The EdU incorporation assay was analyzed by confocal microscopy (N = 3). Scale bar, 50 µm. * *p <* 0.05, ** *p <* 0.01 vs. control.

**Figure 8 ijms-22-10527-f008:**
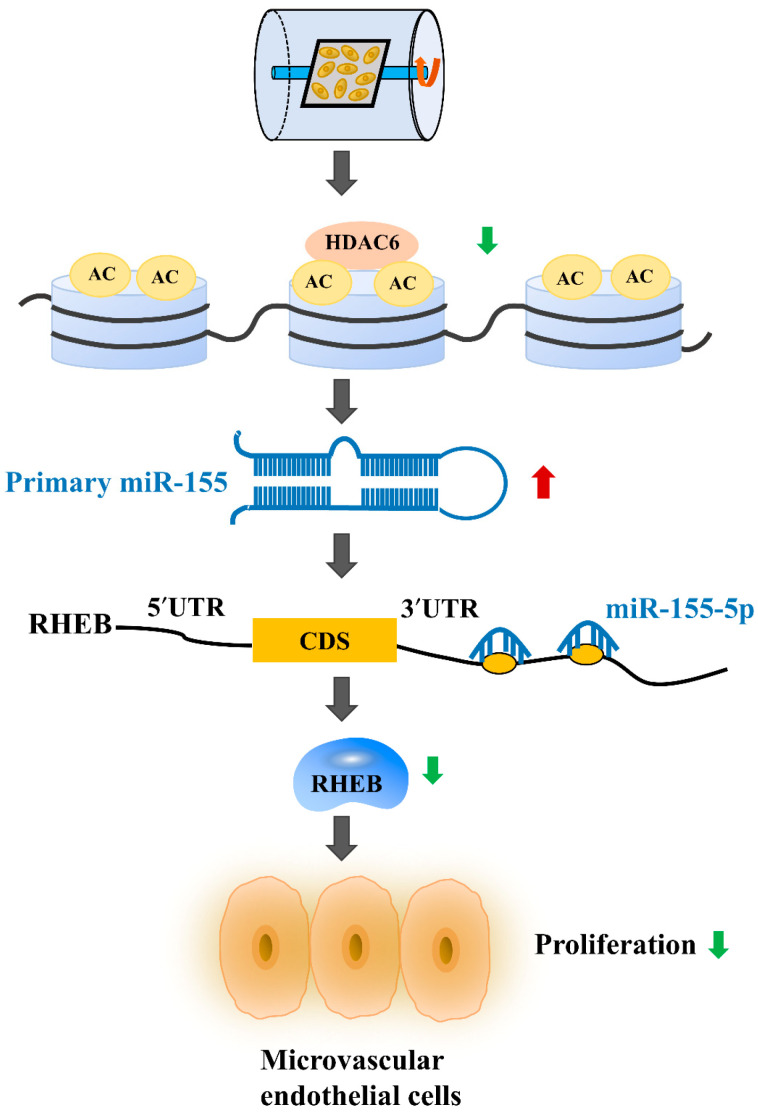
A schematic diagram illustrating the molecular mechanisms by which HDAC6 regulates unloading-induced microvascular endothelial cell growth inhibition. The expression of HDAC6, which is a positive regulator of microvascular endothelial cell proliferation, was downregulated under mechanical unloading. The inhibition of HDAC6 expression promoted miR-155-5p expression, thus inhibiting the expression of RHEB. The green arrow represents suppression and the red arrow represents promotion.

## Data Availability

Not applicable.

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
