# Peer review of "HDAC6 Negatively Regulates miR-155-5p Expression to Elicit Proliferation by Targeting RHEB in Microvascular Endothelial Cells under Mechanical Unloading"

_ijms, 2021, doi:10.3390/ijms221910527_

Round 1

Reviewer 1 Report

  1. In figure 1, GAPDH is not an ideal internal control. Molecular weight is overlapped with the target protein. A new internal control should be performed.

  1. Figure 1A and 1E. the differences between control and clino appear smaller on the blots than they appear on the graphs. The error bars (SD) do not seem to reflect the rather large variance seen in the blots for certain groups.

  1. I found the attached figures to be small and not clear. I attempted to zoom in to be able to read it, however the low resolution of the figures was not helping. Please redo the figures with higher resolution.

  1. I cant review the article before replacement of higher resolution figures

Author Response

Dear reviewer,

Thank you very much for your comments on our submitted manuscript titled “HDAC6 negatively regulates miR-155-5p expression to elicit proliferation by targeting RHEB in microvascular endothelial cells under mechanical unloading” (ijms-1370275). Your suggestions are very constructive, we have carefully revised all of the points that were raised in the revised manuscript and you will see from our detailed answers below. Furthermore, in order to ensure our writing in English syntax, we polished and modified the language of the manuscript again, and we had this manuscript copyedited by a professional English editing service that specializes in scientific papers (American Journal Experts), and the certificate is also submitted. We would like to re-submit it for your consideration. We hope that the revised version of the manuscript is now acceptable for publication in your journal.

Point 1: In figure 1, GAPDH is not an ideal internal control. Molecular weight is overlapped with the target protein. A new internal control should be performed.

Response 1: Thank you very much for your kind comments and valuable suggestions. It has been reported that mechanical unloading or microgravity affects the expression of cytoskeleton proteins. Therefore, the expression of usual internal controls, such as Actin and Tubulin which are cytoskeleton proteins, would change under weightlessness condition[1-3]. However, GAPDH does not change under weightlessness conditions. The studies related to mechanical unloading or microgravity use GAPDH as an internal control[4-7]. Furthermore, HDAC6 affects the acetylation of the cytoskeleton protein, thus most studies on HDAC6 use GAPDH as an internal control[8-10].

In addition, CCK8 and EDU detection are the main methods for detecting cell proliferation, and the conclusions related cell proliferation have been fully verified through these two experiments. PCNA, as a proliferation-related protein, plays a complementary role in evaluating cell proliferation. Considering that the molecular weight of GAPDH is overlapped with the PCNA, we strictly maintain the same grouping and sample amount in different lanes to obtain GAPDH and PCNA at the same time. We could strengthen our conclusions by adding corresponding experiments based on your further suggestions.

Point 2: Figure 1A and 1E. the differences between control and clino appear smaller on the blots than they appear on the graphs. The error bars (SD) do not seem to reflect the rather large variance seen in the blots for certain groups.

Response 2: Thank you very much for your kind comments and valuable suggestions. We have carefully reviewed the figure and histogram. Since the histogram reflects the results of statistical analysis for repeated three times, it may differ slightly from the strip shown. According to your comments, we have replaced the strips in the image (from three repeats) to make them fit the histogram better in Figure 1A and 1E.

In addition, we used Image J software for gray value analysis and Graphpad for drawing. The three repeated strips, gray values and statistical data are shown. (Please see the attachment response letter). Furthermore, according to your suggestion, we have replaced strips of other figures (from three repeats) in the manuscript to make them fit the histogram better, and all the three repeats of the strips have been uploaded to the journal.

Point 3: I found the attached figures to be small and not clear. I attempted to zoom in to be able to read it, however the low resolution of the figures was not helping. Please redo the figures with higher resolution.

Response 3: Thanks for your constructive comments. We are so sorry for your troubles due to our lack of experience. We have uploaded the figures to the journal separately at the time of submission. According to your and another reviewer comments, we have revised and redone the figures with higher resolution. Moreover, we have attached clearer figures in the manuscript, and uploaded the figures as a PDF to you.

Point 4: I can’t review the article before replacement of higher resolution figures

Response 4: Thank you very much for your kind comments and valuable suggestions. We are so sorry for your troubles due to our lack of experience. According to your comments, we have redone the figures with higher resolution and replaced clearer figures in the manuscript.

Special thanks to you for your valuable comments again.

I look forward to hearing from you soon.

With best wishes,

Yours sincerely,

Shu Zhang

Reviewer 2 Report

The topic seems to be very interesting. Unfortunately, this is impossible to honestly review this manuscript due to the poor quality of data presentation. The titles of axes, labels, and legends in most figures are unreadable because of too small font-size. Even magnification up to 200% does not allow to read the values and descriptions in some figures. Moreover, fluorescence images are too small. In addition, the introduction section does not provide enough background.

Author Response

Dear reviewer,

Thank you very much for your comments on our submitted manuscript titled “HDAC6 negatively regulates miR-155-5p expression to elicit proliferation by targeting RHEB in microvascular endothelial cells under mechanical unloading” (ijms-1370275). Your suggestions are very constructive, we have carefully revised all of the points that were raised in the revised manuscript and you will see from our detailed answers below. Furthermore, in order to ensure our writing in English syntax, we polished and modified the language of the manuscript again, and we had this manuscript copyedited by a professional English editing service that specializes in scientific papers (American Journal Experts), and the certificate is also submitted. We would like to re-submit it for your consideration. We hope that the revised version of the manuscript is now acceptable for publication in your journal.

Point 1: The topic seems to be very interesting. Unfortunately, this is impossible to honestly review this manuscript due to the poor quality of data presentation. The titles of axes, labels, and legends in most figures are unreadable because of too small font-size. Even magnification up to 200% does not allow to read the values and descriptions in some figures. Moreover, fluorescence images are too small. In addition, the introduction section does not provide enough background.

Response 1: Thank you very much for your kind comments and valuable suggestions. We are so sorry for your troubles due to our lack of submission experience. We have uploaded the figures to the journal separately at the time of submission. According to your and another reviewer comments, we have revised and redone the figures with higher resolution. Moreover, we have attached clearer figures in the manuscript, and uploaded the figures as a PDF.

Due to typesetting limitations, the fluorescence images are too small, so we have adjusted and attached higher resolution fluorescence images and uploaded clearer fluorescence images as a PDF. In addition, we added background in the introduction and marked red in the revised manuscript. Finally, in order to ensure our writing in English syntax, we polished and modified the language of the manuscript again, and we had this manuscript copyedited by a professional English editing service that specializes in scientific papers (American Journal Experts), and the certificate is also submitted.

Special thanks to you for your valuable comments again.

I look forward to hearing from you soon.

With best wishes,

Yours sincerely,

Shu Zhang

Round 2

Reviewer 1 Report

In this work, Xu shows that the HDAC6/miR-155-29 5p/RHEB pathway is a specific target for the preventative treatment of cardiovascular decondition. The novelty of the work is good. There are however some concerns and comments that should be addressed by the authors in order to consider the manuscript for publication.

  1. The abstract and manuscript are roughly prepared with numerous errors in grammar and spelling which need to be corrected. Such as line 133-134, the expression of PCNA was examined by western blotting after clinorotation for 48 h, and the results showed that proliferating cell nuclear antigen (PCNA). Abbreviation goes before the full name. line 184-185, in vitro should be corrected as in vitro.

  1. Figures in the manuscript are still small and not clear. But those are in good condition in supplementary figures. Please redo it.

  1. Authors claim that HDAC6/miR-155-5p/RHEB signaling pathway plats an important role in growth of microvascular endothelial cells. However, protein level of HDAC6 is higher in Clino plus pEX-HDAC6 than in control (figure2F-G). Cell growth doesn’t show similar trends. In addition, HDAC6 location is more important that its level. The key point is the nuclear translocation of HDAC6 rather than total protein level. Based on it, how to hypothize overexpression of HDAC6 would promote cell proliferation?

  1. How many duplicates are in the RNA‐seq results (Figure3)? The figure legend doesn’t show the important information.

Author Response

Dear reviewer,

    Thank you very much for your comments on our submitted manuscript titled “HDAC6 negatively regulates miR-155-5p expression to elicit proliferation by targeting RHEB in microvascular endothelial cells under mechanical unloading” (ijms-1370275). Your suggestions are very constructive, we have carefully revised all of the points that were raised in the revised manuscript and you will see from our detailed answers below. Furthermore, we have redone the figures, and the figures and fluorescence images have been uploaded as PDFs in supplementary materials. Finally, according to the kind suggestions of you, we polished and modified the language of the manuscript again, and we had this manuscript copyedited by MDPI, and the certificate is also submitted. We would like to re-submit it for your consideration. We hope that the revised version of the manuscript is now acceptable for publication in your journal.

Point 1: The abstract and manuscript are roughly prepared with numerous errors in grammar and spelling which need to be corrected. Such as line 133-134, the expression of PCNA was examined by western blotting after clinorotation for 48 h, and the results showed that proliferating cell nuclear antigen (PCNA). Abbreviation goes before the full name. line 184-185, in vitro should be corrected as in vitro.

Response 1: Thank you very much for your kind comments and valuable suggestions. We have carefully revised grammatical and spelling mistakes word by word and have corrected the errors according to your comment. Please see line 165-167; line 213, 220, 415, 471. Moreover, the revised manuscript has been edited by MDPI and the certificate is also submitted.

Point 2: Figures in the manuscript are still small and not clear. But those are in good condition in supplementary figures. Please redo it.

Response 2: Thanks for your constructive comments. According to your and another reviewer comments, we have redone the figures with higher resolution and larger fonts, and replaced clearer figures in the revised manuscript. Considering that typesetting limitations, we have made appropriate adjustments to figures and put the histograms of Western blotting into the supplementary materials. Moreover, we have uploaded the figures as a PDF in the supplementary materials.

Point 3: Authors claim that HDAC6/miR-155-5p/RHEB signaling pathway plats an important role in growth of microvascular endothelial cells. However, protein level of HDAC6 is higher in Clino plus pEX-HDAC6 than in control (figure2F-G). Cell growth doesn’t show similar trends. In addition, HDAC6 location is more important that its level. The key point is the nuclear translocation of HDAC6 rather than total protein level. Based on it, how to hypothize overexpression of HDAC6 would promote cell proliferation?

Response 3: Thanks for your constructive comments. Mechanical unloading affects cell proliferation through multiple factors and signaling pathways, among which HDAC6 is one. Our results showed that overexpression of HDAC6 could partially rescue the inhibition of proliferation induced by unloading, but could not completely reverse it. In addition, overexpression HDAC6 may exist saturation. Therefore, although protein level of HDAC6 is higher in Clino plus pEX-HDAC6 than in control, cell growth is higher in control compared with Clino plus pEX-HDAC6 group.

According to your valuable suggestions, we further supplemented our experiments, and the experimental results are as follows. We first found the RNA and protein expression of HDAC6 decreased under unloading (Figure A and B) and further conducted nucleo-cytoplasmic separation experiments on the Con and Clino group. The result showed that the expression of HDAC6 decreased both in the nucleus and cytoplasm under unloading (Figure C). Furthermore, the nucleo-cytoplasmic separation experiments showed that HDAC6 increased significantly in Clino plus pEX-HDAC6 group, indicating that unloading might not significantly affect nuclear translocation (Figure D). All these results suggested that the change of HDAC6 protein level was more significant rather than nuclear translocation under unloading. Furthermore, we have reviewed relevant literature and found that HDAC6 expression change also plays an important role in regulating cell function[1, 2]. Therefore, we pay more attention to the changes of HDAC6 protein level. Based on them, we examined overexpression of HDAC6 would promote cell proliferation through transfection.

References:

  1. Yang, H. D.; Kim, H. S.; Kim, S. Y.; Na, M. J.; Yang, G.; Eun, J. W.; Wang, H. J.; Cheong, J. Y.; Park, W. S.; Nam, S. W., HDAC6 Suppresses Let-7i-5p to Elicit TSP1/CD47-Mediated Anti-Tumorigenesis and Phagocytosis of Hepatocellular Carcinoma. Hepatology (Baltimore, Md.) 2019, 70, (4), 1262-1279.
  2. Sun, N.; Wang, C. Y.; Sun, Y. Q.; Ruan, Y. J.; Huang, Y. Y.; Su, T.; Zhou, X. H.; Huang, H.; Guo, W. J.; He, M. Q.; Yao, R. X.; Lin, X. J., Down-regulated miR-148b increases resistance to CHOP in diffuse large B-cell lymphoma cells by rescuing Ezrin. Biomedicine & pharmacotherapy = Biomedecine & pharmacotherapie 2018, 106, 267-274.

Point 4: How many duplicates are in the RNA‐seq results (Figure3)? The figure legend doesn’t show the important information.

Response 4: Thanks for your kind comments. There are three duplicates in the RNA‐seq results, and we have added the important information “N=3” in the figure legend. Please see line 276.

Special thanks to you for your valuable comments again.

I look forward to hearing from you soon.

With best wishes,

Yours sincerely,

Shu Zhang

Reviewer 2 Report

The authors investigated the role of HDAC6/miR-155-5p/RHEB pathway in the inhibition of the brain microvascular endothelial cells proliferation. The subject is very interesting, however, several concerns arose regarding the described research.

The introduction section does not provide enough background. For example, since long-term exposure to microgravity and prolonged bed rest are still not common, thus the authors should briefly define the following terms: mechanical unloading, cardiovascular deconditioning (meaning and symptoms) and bed rest. These are interesting issues, but still not applicable for most of the potential readers, who are not faced with the conditions of weightlessness. In addition, the physiological relevance is not clear. The most important question is why the authors decided to analyse a brain-derived endothelial cell proliferation. Is it because of angiogenesis? However, the authors do not elucidate the importance of angiogenesis in the brain during long-term exposure to microgravity and/or cardiovascular deconditioning. Moreover, the authors cite several reports regarding the involvement of the analysed proteins and microRNAs in diverse diseases (mostly in respect to cancer cells), but do not mention why the studied mechanism is important for microvascular endothelial cells. Why the effects of unloading and HDAC6 on endothelial cells proliferation are important? All such information should be briefly described in the Introduction section to provide a background for the described results and discussion.

The authors did not change the descriptions of axes and legends in figures. This is not only the matter of resolution, but the size of fonts. The worst situation is in Fig. 3. Please, try to print the figures in 100% scale and try to read the labels in Fig. 3B-C. It is completely impossible. Another example is Fig. 3H - what is placed over the blue box described as a ‘Binding site’? It is unreadable. Other fonts are also too small, but after enlargement they are possible to read. However, readers should have an opportunity to read the descriptions in graphs in printed form of publication, and not only after enlargement of the scale to 300%. The data absolutely should be made readable. The easiest way to verify whether the font is large enough is to print the panel of figures and try to read it. In the present form it is impossible. The individual graphs in the panels should have increased font size and set again into the panels. There is a lot of space for larger fonts.

The authors write (lines 134-136) that ‘proliferating cell nuclear antigen (PCNA) expression was significantly downregulated during unloading, which was accompanied by the inhibition of microvascular formation (Figure 1A).’ What does ‘microvascular formation’ mean? There is not presented any formation of microvasculature. Please explain this. Furthermore (lines 138-140): “The cell activity continuously increased in a time-dependent manner, while that in the con group increased more obviously, indicating that mechanical unloading inhibited microvascular endothelial cell proliferation (Figure 1B).” What does ‘cell activity’ mean? Is it the rate of proliferation? ‘Cell activity’ is not a precise term.

There are some concerns regarding the research design. The authors do not explain the reason to select Ac-H3K9 for further analysis.

The authors do not deliver any functional data, which could demonstrate the importance of the described studies.

Please, explain why optical density values exceeding 1 are used. Does ‘optical density’ mean ‘absorbance’? It should not exceed 1.

Author Response

Dear reviewer,

Thank you very much for your comments on our submitted manuscript titled “HDAC6 negatively regulates miR-155-5p expression to elicit proliferation by targeting RHEB in microvascular endothelial cells under mechanical unloading” (ijms-1370275). Your suggestions are very constructive, we have carefully revised all of the points that were raised in the revised manuscript and you will see from our detailed answers below. Furthermore, we have redone the figures, and the figures and fluorescence images have been uploaded as PDFs in supplementary materials. Finally, according to the kind suggestions of you, we polished and modified the language of the manuscript again, and we had this manuscript copyedited by MDPI, and the certificate is also submitted. We would like to re-submit it for your consideration. We hope that the revised version of the manuscript is now acceptable for publication in your journal.

Point 1: The introduction section does not provide enough background. For example, since long-term exposure to microgravity and prolonged bed rest are still not common, thus the authors should briefly define the following terms: mechanical unloading, cardiovascular deconditioning (meaning and symptoms) and bed rest. These are interesting issues, but still not applicable for most of the potential readers, who are not faced with the conditions of weightlessness. In addition, the physiological relevance is not clear. The most important question is why the authors decided to analyse a brain-derived endothelial cell proliferation. Is it because of angiogenesis? However, the authors do not elucidate the importance of angiogenesis in the brain during long-term exposure to microgravity and/or cardiovascular deconditioning. Moreover, the authors cite several reports regarding the involvement of the analysed proteins and microRNAs in diverse diseases (mostly in respect to cancer cells), but do not mention why the studied mechanism is important for microvascular endothelial cells. Why the effects of unloading and HDAC6 on endothelial cells proliferation are important? All such information should be briefly described in the Introduction section to provide a background for the described results and discussion.

Response 1: Thanks for your kind comments and valuable suggestions. Based on your constructive comment, we have revised and added background in the Introduction section and marked red in the revised manuscript. Firstly, we have briefly defined the following terms: mechanical unloading, cardiovascular deconditioning and bed rest. Furthermore, we explain the reason why we choose bEnd.3 to analyze microvascular endothelial cell proliferation. ECs play important roles in regulating the function of blood vessels and maintaining the stability of the cardiovascular system[1, 2]. Mechanical unloading means decreased physical activity with reduced hemodynamic activity and local vascular shear stress, which can result in changes in the morphology and ultrastructure of vascular endothelial cells, as well as the functions of secretion, proliferation, apoptosis and angiogenesis[3, 4]. bEnd.3 cell is an internationally recognized immortalized and stable microvascular endothelial cell line with various characteristics of microvascular endothelial cells. Moreover, bEnd.3 cell derives from mice commonly used in experiments, it is easy to perform the studies in vivo [5-7]. Taken together, bEnd.3 cell is an ideal model for studying the functions of microvascular endothelial cells in vitro. Please see line 48-73.

In addition, we explain the importance of unloading and HDAC6 on endothelial cells proliferation. Please see line. Recent studies have shown that HDAC6 can serve as a crucial regulator in chromatin maintenance and function by regulating histone acetylation, thus regulating mRNA gene expression and miRNAs expression[8-10]. These studies provide an insight as to whether HDAC6 acts as an effective histone modifier to regulate functional miRNAs, especially mechanoresponsive microRNAs, thereby regulating the function of microvascular endothelial cells under mechanical unloading. Our study further found that the expression of HDAC6 decreased under unloading, especially in the nucleus, and then verified whether HDAC6 regulated the proliferation of microvascular endothelial cells through gain and loss-functional studies and CHIP assays. Please see line 96-108, 119-122.

Point 2: The authors did not change the descriptions of axes and legends in figures. This is not only the matter of resolution, but the size of fonts. The worst situation is in Fig. 3. Please, try to print the figures in 100% scale and try to read the labels in Fig. 3B-C. It is completely impossible. Another example is Fig. 3H - what is placed over the blue box described as a ‘Binding site’? It is unreadable. Other fonts are also too small, but after enlargement they are possible to read. However, readers should have an opportunity to read the descriptions in graphs in printed form of publication, and not only after enlargement of the scale to 300%. The data absolutely should be made readable. The easiest way to verify whether the font is large enough is to print the panel of figures and try to read it. In the present form it is impossible. The individual graphs in the panels should have increased font size and set again into the panels. There is a lot of space for larger fonts.

Response 2: Thanks for your constructive comments. According to your valuable suggestions, we have redone the figures with higher resolution and larger fonts, and replaced clearer figures in the revised manuscript. Considering that typesetting limitations, we have made appropriate adjustments to figures and put the histograms of Western blotting into the supplementary materials. Moreover, we have uploaded the figures as a PDF in the supplementary materials.

Point 3: The authors write (lines 134-136) that ‘proliferating cell nuclear antigen (PCNA) expression was significantly downregulated during unloading, which was accompanied by the inhibition of microvascular formation (Figure 1A).’ What does ‘microvascular formation’ mean? There is not presented any formation of microvasculature. Please explain this. Furthermore (lines 138-140): “The cell activity continuously increased in a time-dependent manner, while that in the con group increased more obviously, indicating that mechanical unloading inhibited microvascular endothelial cell proliferation (Figure 1B).” What does ‘cell activity’ mean? Is it the rate of proliferation? ‘Cell activity’ is not a precise term.

Response 3: Thanks for your constructive comments. We have carefully reviewed the manuscript. The article focuses on cell proliferation which is the potent process of angiogenesis. We are so sorry for your troubles due to our errors and we have deleted and revised these issues you have mentioned. Furthermore, cell activity means the rate of cell growth, and we have revised it in the manuscript. Please see line168, 171.

Point 4: There are some concerns regarding the research design. The authors do not explain the reason to select Ac-H3K9 for further analysis.

Response 4: Thanks for your kind comments. Based on your advice, we have added the reason to select Ac-H3K9 for further analysis. It is well known that HDAC6 can serve as a crucial regulator in chromatin maintenance and function by regulating histone acetylation. Recent studies showed that HDAC6 could deacetylate H3K9, and H3K9 deacetylation at the promoter usually results in reduced transcription[8, 11, 12]. Therefore, we performed ChIP assays with Ac-H3K9 in the context of HDAC6 knockdown in bEnd.3 cells. Results indicated that Ac-H3K9 was enriched in the sites of the miR-155-5p promoter. Moreover, knockdown of HDAC6 significantly increased H3K9 acetylation in the miR-155-5p promoter regions in bEnd.3 cells. Please see line 260-265.

Point 5: The authors do not deliver any functional data, which could demonstrate the importance of the described studies.

Response 5: Thanks for your kind comments and valuable suggestions. Mechanical unloading can result in changes in the morphology and ultrastructure of vascular endothelial cells, as well as the functions of secretion, proliferation, apoptosis and angiogenesis. Cell proliferation is the potent process of angiogenesis and cardiovascular deconditioning. The article focuses on cell proliferation, and we have shown proliferation related functional data, including CCK-8 and EDU. According to your kind suggestions, we are conducting other functional experiments such as wound healing and tubule formation. Because there are numbers of groups involved, we will need more time if it is needed.

Point 6: Please, explain why optical density values exceeding 1 are used. Does ‘optical density’ mean ‘absorbance’? It should not exceed 1.

Response 6: Thanks for your constructive comments. Optical density means absorbance. Optical density values were measured at 450 nm through the microplate reader. According to your valuable suggestions, we have reviewed the high-quality literature related to CCK-8 assays and repeated our experiments, which support that optical density could exceed 1 [13-16] . If you still have questions, we could discuss this issue further.

References

  1. Maier, J. A.; Cialdai, F., The impact of microgravity and hypergravity on endothelial cells. BioMed research international 2015, 2015, 434803.
  2. Cines, D. B.; Pollak, E. S.; Buck, C. A.; Loscalzo, J.; Zimmerman, G. A.; McEver, R. P.; Pober, J. S.; Wick, T. M.; Konkle, B. A.; Schwartz, B. S.; Barnathan, E. S.; McCrae, K. R.; Hug, B. A.; Schmidt, A. M.; Stern, D. M., Endothelial cells in physiology and in the pathophysiology of vascular disorders. Blood 1998, 91, (10), 3527-61.
  3. Li, C. F.; Sun, J. X.; Gao, Y.; Shi, F.; Pan, Y. K.; Wang, Y. C.; Sun, X. Q., Clinorotation-induced autophagy via HDM2-p53-mTOR pathway enhances cell migration in vascular endothelial cells. Cell death & disease 2018, 9, (2), 147.
  4. Li, C. F.; Pan, Y. K.; Gao, Y.; Shi, F.; Wang, Y. C.; Sun, X. Q., Autophagy protects HUVECs against ER stress-mediated apoptosis under simulated microgravity. Apoptosis: an international journal on programmed cell death 2019, 24, (9-10), 812-825.
  5. Oh, Y. S.; Choi, M. H.; Shin, J. I., Co-Culturing of Endothelial and Cancer Cells in a Nanofibrous Scaffold-Based Two-Layer System. International journal of molecular sciences 2020, 21, (11).
  6. Cui, P. H.; Petrovic, N.; Murray, M., The ω-3 epoxide of eicosapentaenoic acid inhibits endothelial cell proliferation by p38 MAP kinase activation and cyclin D1/CDK4 down-regulation. British journal of pharmacology 2011, 162, (5), 1143-55.
  7. 乐飞, 张国平, 殷莲华, 小鼠脑微血管内皮细胞株bEnd.3的细胞特征及基因表达谱分析. 中国病理生理杂志 2004, 20, (8), 1340-1344.
  8. Kai, H.; Wu, Q.; Yin, R.; Tang, X.; Shi, H.; Wang, T.; Zhang, M.; Pan, C., LncRNA NORAD Promotes Vascular Endothelial Cell Injury and Atherosclerosis Through Suppressing VEGF Gene Transcription via Enhancing H3K9 Deacetylation by Recruiting HDAC6. Frontiers in cell and developmental biology 2021, 9, 701628.
  9. Li, C.; Zhou, Y.; Loberg, A.; Tahara, S. M.; Malik, P.; Kalra, V. K., Activated Transcription Factor 3 in Association with Histone Deacetylase 6 Negatively Regulates MicroRNA 199a2 Transcription by Chromatin Remodeling and Reduces Endothelin-1 Expression. Molecular and cellular biology 2016, 36, (22), 2838-2854.
  10. Jia, Y. J.; Liu, Z. B.; Wang, W. G.; Sun, C. B.; Wei, P.; Yang, Y. L.; You, M. J.; Yu, B. H.; Li, X. Q.; Zhou, X. Y., HDAC6 regulates microRNA-27b that suppresses proliferation, promotes apoptosis and target MET in diffuse large B-cell lymphoma. Leukemia 2018, 32, (3), 703-711.
  11. Yang, H. D.; Kim, H. S.; Kim, S. Y.; Na, M. J.; Yang, G.; Eun, J. W.; Wang, H. J.; Cheong, J. Y.; Park, W. S.; Nam, S. W., HDAC6 Suppresses Let-7i-5p to Elicit TSP1/CD47-Mediated Anti-Tumorigenesis and Phagocytosis of Hepatocellular Carcinoma. Hepatology (Baltimore, Md.) 2019, 70, (4), 1262-1279.
  12. Festa Ortega, J. F.; Heidor, R.; Auriemo, A. P.; Marques Affonso, J.; Pereira D' Amico, T.; Herz, C.; de Conti, A.; Ract, J.; Gioieli, L. A.; Purgatto, E.; Lamy, E.; I, P. P.; Salvador Moreno, F., Butyrate-containing structured lipids act on HDAC4, HDAC6, DNA damage and telomerase activity during promotion of experimental hepatocarcinogenesis. Carcinogenesis 2021, 42, (8), 1026-1036.
  13. Zhang, L.; Cheng, H.; Yue, Y.; Li, S.; Zhang, D.; He, R., H19 knockdown suppresses proliferation and induces apoptosis by regulating miR-148b/WNT/β-catenin in ox-LDL -stimulated vascular smooth muscle cells. Journal of biomedical science 2018, 25, (1), 11.
  14. Ma, Z.; Liu, D.; Di, S.; Zhang, Z.; Li, W.; Zhang, J.; Xu, L.; Guo, K.; Zhu, Y.; Li, X.; Han, J.; Yan, X., Histone deacetylase 9 downregulation decreases tumor growth and promotes apoptosis in non-small cell lung cancer after melatonin treatment. Journal of pineal research 2019, 67, (2), e12587.
  15. Zhang, Z.; Li, J.; Huang, Y.; Peng, W.; Qian, W.; Gu, J.; Wang, Q.; Hu, T.; Ji, D.; Ji, B.; Zhang, Y.; Wang, S.; Sun, Y., Upregulated miR-1258 regulates cell cycle and inhibits cell proliferation by directly targeting E2F8 in CRC. Cell proliferation 2018, 51, (6), e12505.
  16. Liao, L.; Ge, M.; Zhan, Q.; Huang, R.; Ji, X.; Liang, X.; Zhou, X., PSPH Mediates the Metastasis and Proliferation of Non-small Cell Lung Cancer through MAPK Signaling Pathways. International journal of biological sciences 2019, 15, (1), 183-194.

Special thanks to you for your valuable comments again.

I look forward to hearing from you soon.

With best wishes,

Yours sincerely,

Shu Zhang

Round 3

Reviewer 2 Report

I recommend to accept the manuscript for publication.

One comment regarding the absorbance exciding 1.0 - this is related to the assumptions of the Lambert-Beer’s law. The sample should not be too concentrated – each molecule should have the same probability to accept a photon and should not interact with other molecules. However, some new spectrophotometers have possibility to corrects such effects. Some scientists present absorbance values exciding 1, but very often incorrectly. The simplest way to avoid uncertainty is to dilute samples.